# Strawberry, Blueberry, and Strawberry-Blueberry Blend Beverages Prevent Hepatic Steatosis in Obese Rats by Modulating Key Genes Involved in Lipid Metabolism

**DOI:** 10.3390/ijerph20054418

**Published:** 2023-03-01

**Authors:** Ana María Sotelo-González, Rosalía Reynoso-Camacho, Ana Karina Hernández-Calvillo, Ana Paola Castañón-Servín, David Gustavo García-Gutiérrez, Haiku Daniel de Jesús Gómez-Velázquez, Miguel Ángel Martínez-Maldonado, Ericka Alejandra de los Ríos, Iza Fernanda Pérez-Ramírez

**Affiliations:** 1Chemistry School, Universidad Autónoma de Querétaro, Querétaro 76010, Mexico; 2Unidad Académica Multidisciplinaria Reynosa-Aztlán, Universidad Autónoma de Tamaulipas, Reynosa 88740, Mexico; 3Facultad de Estudios Superiores Cuautilán, Universidad Nacional Autónoma de México, Querétaro 76231, Mexico; 4División de Gastronomía, Instituto Tecnológico Superior de Huichapan, Huichapan 42411, Mexico; 5Institute of Neurobiology, Campus Juriquilla, Universidad Nacional Autónoma de México, Querétaro 76230, Mexico

**Keywords:** berry fruits, functional beverages, obesity, hepatic steatosis, rats, urinary polyphenols

## Abstract

There is an increasing interest in developing natural herb-infused functional beverages with health benefits; therefore, in this study, we aimed to evaluate the effect of strawberry, blueberry, and strawberry-blueberry blend decoction-based functional beverages on obesity-related metabolic alterations in high-fat and high-fructose diet-fed rats. The administration of the three berry-based beverages for eighteen weeks prevented the development of hypertriglyceridemia in obese rats (1.29–1.78-fold) and hepatic triglyceride accumulation (1.38–1.61-fold), preventing the development of hepatic steatosis. Furthermore, all beverages significantly down-regulated *Fasn* hepatic expression, whereas the strawberry beverage showed the greatest down-regulation of *Acaca*, involved in fatty acid de novo synthesis. Moreover, the strawberry beverage showed the most significant up-regulation of hepatic *Cpt1* and *Acadm* (fatty acid β-oxidation). In contrast, the blueberry beverage showed the most significant down-regulation of hepatic *Fatp5* and *Cd36* (fatty acid intracellular transport). Nevertheless, no beneficial effect was observed on biometric measurements, adipose tissue composition, and insulin resistance. On the other hand, several urolithins and their derivatives, and other urinary polyphenol metabolites were identified after the strawberry-based beverages supplementation. In contrast, enterolactone was found significantly increase after the intake of blueberry-based beverages. These results demonstrate that functional beverages elaborated with berry fruits prevent diet-induced hypertriglyceridemia and hepatic steatosis by modulating critical genes involved in fatty acid hepatic metabolism.

## 1. Introduction

The global market of functional beverages was about $117 billion USD in 2021 and is expected to grow to $156 billion USD in 2026 at a compound annual growth rate of 6%. The key reasons associated with the growth in the functional beverages market include increased awareness of health-conscious consumers shifting from hypercaloric juices or carbonated beverages to healthier hydration products. Moreover, consumers are interested in ready-to-drink functional beverages with clean labels, including low-calorie natural sweeteners and natural pigments [1,2].

Interestingly, the natural botanical-infused drinks trend dominated the functional beverage market in 2021, including using plant extracts, such as roots, flowers, leaves, seeds, and fruits [2]. In this regard, berry fruits have been widely used to develop functional beverages due to their sensory attributes and bioactive compound content [3].

We previously demonstrated that strawberry and blueberry decoctions are rich in polyphenols, mainly anthocyanins; nevertheless, these decoctions showed contrasting polyphenol profiles. The strawberry decoction was rich in pelargonidin hexoside and included ellagitannins as minor components. Conversely, the blueberry decoction showed a high content of petunidin hexoside, malvidin hexoside, delphinidin hexoside, and cyanidin hexoside, as several flavonols as minor components [4].

Polyphenols can be absorbed in the small intestine or reach the colon, which can be metabolized by colonic microbiota and absorbed in the large intestine. Once absorbed, polyphenols suffer phase II metabolism (glucuronidation, sulphation, and methylation) in enterocytes and hepatocytes, which are then distributed to their target organs and are further excreted in urine [5]. Several anthocyanin metabolites, mainly glucuronide derivatives, have been reported associated with consuming blueberry and strawberry fruit and juice. In addition, ellagic acid and urolithins have been identified after strawberry fruit intake, produced during the metabolism of ellagitannins.

Interestingly, several studies have demonstrated that berry fruits exert beneficial effects mainly related to improving cardiovascular health. It has been previously demonstrated that strawberry and blueberry fruit prevent the development of hepatic steatosis hepatic by decreasing lipid accumulation, improving insulin sensitivity, and decreasing inflammation and oxidative stress based on in vitro, in vivo, and clinical studies [6]. Moreover, it has been demonstrated that a blended powder elaborated with freeze-dried strawberry and blueberry also exerts these insulin sensitizers and hypolipidemic effects, attributed to the modulation of inflammatory and lipogenic biomarkers [7]. Regarding berry-based beverages, the supplementation of a strawberry smoothie blended with other fruits ameliorated hepatic steatosis in high-fat diet-fed obese mice [8]. Similarly, blueberry juices ameliorate the development of prediabetes and hepatic steatosis in high-sucrose and high-fat diet-fed rats [9]. These health benefits have been attributed to their high content and diversity of polyphenols when different whole fruits, juices, or smoothie-based beverages were used.

Interestingly, we have previously demonstrated that strawberry and blueberry decoctions are rich in polyphenols showing potential complementary profiles, and thus can be proposed as ingredients for the elaboration of functional beverages; in addition, these beverages are low in calories [10]. Therefore, to contribute to developing natural botanical-infused functional beverages with health benefits, this study aimed to evaluate the preventive effect of polyphenol-rich strawberry and blueberry decoction-based beverages on developing obesity-related metabolic alterations in high-fat and high-fructose (HFF) diet-fed rats. Moreover, phase II and colonic polyphenol urinary metabolites were identified as potential contributors to the health-beneficial effects of berry-based beverages.

## 2. Materials and Methods

### 2.1. Chemical and Reagents

Optima LC/MS grade methanol and water; and JT Baker potassium sorbate, sodium carbonate, Invitrogen Trizol reagent; and ACS grade chloroform, methanol, and ethanol were purchased from Fisher Scientific (Waltham, MA, USA). Citric acid, 1 N Folin–Ciocalteu reagent, 99% formic acid, (−)-epicatechin, naringenin, quercetin, cyanidin chloride, ellagic acid, gallic acid, 4-hydroxyphenylpropionic acid, 4-hydroxyphenylacetic acid, enterodiol, hippuric acid, 2,2-diphenyl-1-picrylhydrazyl (DPPH), 2,2′-azino-bis(3-ethylbenzo-thiazoline-6-sulfonic acid) (ABTS), 6-hydroxy-2,5,7,8-tetramethylchroman-2-carboxylic acid (Trolox), potassium persulphate were purchased from Sigma-Aldrich (St. Louis, MO, USA). M-MLV RT, M-MLV 5× reaction buffer, oligo (dT)15 primer, dNTP mix, SybrGreen master mix, and RNAse-free water were purchased from Promega.

### 2.2. Elaboration of the Berry-Based Beverages

Strawberry and blueberry decoctions (aqueous extracts) were elaborated at 10% (*w*/*v*) at 95 °C for 15 min. Then, beverages were prepared by mixing the berry decoctions (100% strawberry, 100% blueberry, and 50% strawberry/50% blueberry, respectively) with a preservative (potassium sorbate), non-caloric sweeteners (sucralose and acesulfame K), and a pH regulator (citric acid). Finally, the berry-based beverages were pasteurized at 73 °C for 15 s and immediately cooled in ice. All beverages were stored at 4 °C for 5 days until use. Microbiologic analyses (total aerobic mesophyll bacteria count, total coliform count, fecal coliform count, and mold and yeast count) were assessed to guarantee to be innocuous for the in vivo study.

### 2.3. Polyphenol Profile by UPLC-Q-ToF MS

Freshly prepared beverages were passed through syringe PVDF filters (0.45 μm, 13 mm), and were analyzed in an Ultra-Performance Liquid Chromatograph (UPLC) coupled to a Quadrupole/Time-of-Flight Mass Spectrometer (Q-ToF MS) with an electrospray ionization (ESI) interphase (Vion, Waters Co., Milford, MA, USA). Samples (1 μL) were injected into a BEH Acquity C18 column (2.1 × 100 mm^2^, 1.7 μm) at 35 °C. The mobile phase consisted of (A) water:formic acid (99:1 *v*/*v*), and (B) acetonitrile:formic acid (99:1 *v*/*v*). Gradient conditions and Q-Tof MS processing and acquisition conditions were previously reported by Reynoso-Camacho et al. [4]. Mass spectra were analyzed for the identification of polyphenols by comparison of their exact mass (mass error < 5 ppm) and fragmentation patterns. Calibration curves were constructed with (−)-epicatechin (flavanols), naringenin (flavanones), quercetin (flavonols), cyanidin chloride (anthocyanins), ellagic acid (hydroxycinnamic acids and ellagitannins), and gallic acid (hydroxybenzoic acids). High-resolution mass spectra at high and low collision energy of the significant polyphenols identified in this study are shown in Appendix A.

### 2.4. In Vivo Experimental Procedure

The animal experiment was performed following the guidelines of the National Research Council for using experimental animals. It was approved by the Bioethics Committee of the Chemistry School of the Autonomous University of Querétaro (Querétaro, México) (approval number: CBQ16/0831). Fifty male Wistar rats of 160–180 g weight were purchased from the Institute of Neurobiology of the Universidad Nacional Autónoma de México (Querétaro, México). Rats were maintained at 24 ± 1 °C and 50 ± 10% RH under a 12/12 h light/dark cycle.

After one week of acclimation, rats were randomly divided into five experimental groups of ten animals each: (i) standard diet-fed group (Rodent Lab Chow 5001); (ii) HFF diet-fed group (standard diet added with 20% lard and 18% fructose); (iii) HFF diet-fed group supplemented with the strawberry beverage (SB) group; (iv) HFF diet-fed group supplemented with the blueberry beverage (BB) group; and (v) HFF diet-fed group supplemented with the strawberry-blueberry blend beverage (SBB) group.

Berry-based beverages were prepared every five days and were administered for 12 h at night, followed by 12 h of tap water. In contrast, tap water was administered ad libitum to the control groups. Standard or HFF diets were administered ad libitum throughout the experiment, and food and beverage intake was recorded daily throughout the experiment.

After 18 weeks, rats were placed in metabolic cages to recollect of urine and feces. Then, rats were anesthetized with pentobarbital (0.4 mL/kg of body weight via intraperitoneal) and were euthanized by decapitation. Blood was collected into Vacutainer tubes (BD Co., Bergen, NJ, USA) and was centrifuged at 1000× *g* for 10 min for serum separation. Liver and mesenteric, epididymal, and perirenal adipose tissues were collected. A fraction of each liver was stored in 10% neutral buffered formalin (pH 7.4) for histology analysis. The rest of the organs and the biological fluids were snap-frozen in liquid nitrogen and stored at −80 °C until analysis.

#### 2.4.1. Biometric and Adipose Tissue Measurements

Body weight was measured weekly, and biometric measurements were assessed every two weeks. The biometric measurements included abdominal circumference, thoracic circumference, and naso-anal length. With these data, body mass index (BMI), Lee index, and rate of body mass gain were calculated with the following equations, which have been proposed to estimate obesity in rats [10]:(1)$$\mathrm{BMI}=\frac{\mathrm{body}\;\mathrm{weight}\;\left(g\right)}{{\left[\mathrm{length}\;\left(\mathrm{cm}\right)\right]}^{2}}$$
(2)$$\mathrm{Lee}\;\mathrm{index}=\frac{\sqrt[{3}]{\mathrm{bodyweight}\;\left(g\right)\times 10}}{\mathrm{length}\;\left(\mathrm{mm}\right)}$$

The relative weight of mesenteric, epididymal, perirenal, and total adipose tissue was determined. The adiposity index was estimated with the following equation:(3)$$\mathrm{Adiposity}\;\mathrm{index}\;\left(\%\right)=\frac{\mathrm{Total}\;\mathrm{adipose}\;\mathrm{tissue}\;\left(g\right)}{\mathrm{body}\;\mathrm{weight}\;\left(g\right)}*100$$

#### 2.4.2. Determination of Serum Biochemical Analysis

Serum glucose (Glucose-LQ GOD-POD, Spinreact, Girone, Spain), triglycerides (Triglycerides-LQ GPO-POD, Spinreact), aspartate transaminase (GOT-AST-LQ, Spinreact), and alanine transaminase (GPT-ALT-LQ, Spinreact) were determined using commercial enzymatic-colorimetric kits following the manufacturer’s instructions. Following the manufacturers’ instructions, serum insulin was determined using an ELISA kit (EZRMI-13K, EMD Millipore Co., Burlington, MA, USA). Insulin resistance- and beta-Homeostatic model assessment (HOMA-IR and HOMA-beta), quantitative insulin sensitivity check index (QUICKI), fasting glucose-to-insulin ratio (FGIR), and fasting triglycerides and glucose (TyG) indexes were calculated with the following equations:(4)HOMA−IR=glucose (mgdL)×insulin(μMmL)2430
(5)HOMA−Beta=insulin (ngmL)×360glucose (mgdL)−63
(6)QUICKI=1log[insulin(μMmL)]+log[glucose(mgdL)]
(7)FGIR=glucose (mgdL)insulin (μMmL)
(8)TyG=ln[glucose(mgdL)∗triglycerides(mgdL)]2

#### 2.4.3. Triglyceride Extraction and Quantification in Feces, Adipose Tissue, and Liver

Feces were dried at 45 °C for 24 h prior to triglyceride extraction. Dried feces (200 mg) were mixed with 2 mL of 0.9% NaCl and with 2 mL of 2:1 (*v*/*v*) chloroform:methanol. Next, samples were centrifuged at 6000× *g* for 5 min at 25 °C, and 1 mL of the lower phase was collected in a new tube. Then, samples were vacuum dried at 35 °C and resuspended in 200 μL of ethanol. Finally, following the manufacturer’s instructions, triglycerides were determined using a commercial enzymatic-colorimetric kit (Triglycerides-LQ GPO-POD, Spinreact). Results were expressed as mg of triglycerides/g of feces [11]. 

Frozen livers (200 mg) were digested with 350 μL of 30% potassium hydroxide in ethanol (2:1, *v*/*v*) at 55 °C overnight. Then, 650 μL water:ethanol (1:1, *v*/*v*) was added, and samples were centrifuged at 6000× *g* for 5 min. Supernatants (200 μL) were recovered and mixed with 215 μL of 1 M magnesium chloride. Samples were incubated in ice for 10 min and centrifuged at 6000× *g*. Finally, following the manufacturer’s instructions, triglycerides were determined using a commercial enzymatic-colorimetric kit (Triglycerides-LQ GPO-POD, Spinreact). Results were expressed as mg of triglycerides/g of the liver [12].

#### 2.4.4. Fatty Acid Profile in Liver

Frozen livers (50 mg) were extracted with 400 μL of 1.25 M potassium hydroxide in methanol for 60 s and were sonicated for 5 min at 40 kHz at room temperature. Then, samples were mixed with 400 μL of 1.75 M sulfuric acid in methanol for 60 s and sonicated for 5 min at 40 kHz at room temperature. Finally, samples were mixed with 0.8 mL of n-hexane for 30 s, centrifuged at 10,000× *g* for 5 min at 25 °C, and supernatants were recovered [13].

Derivatized samples (1 μL) were injected into an HP-88 capillary column (30 m × 0.25 mm, 0.25 μm) in a Gas Chromatograph (Agilent 7890A, Agilent Technologies Inc., Santa Clara, CA, USA) coupled to a simple-Quadrupole Mass Spectrometer (Agilent 5976C) with an electron impact (EI) ionization source (GC/EI-Q MS). The injector temperature was set at 250 °C in split mode (1:50). Helium was used as carrier gas at 1 mL/min. The oven temperature gradient was set as follows: 50 °C held for 1 min, then raised to 175 °C at 15 °C/min, then raised to 240 °C at 1 °C/min and held for 5 min. The following MS conditions were used: Q MS at 150 °C, EI ionization source at 230 °C, MS electron energy at 70 eV with a mass range of 50–1100 m/z, and a solvent delay of 6.4 min. Results were expressed as mg/g of the liver.

#### 2.4.5. Relative Expression of Genes Involved in Hepatic Lipid Metabolism

RNA was extracted from liver samples (30–50 mg) with Trizol reagent following the manufacturer’s instructions. Then, RNA integrity was confirmed by 0.5% agarose gel electrophoresis, whereas RNA purity (260/280 and 260/230 ratios) and concentration (260 nm) were determined in a microplate spectrophotometer (Multiskan GO, Thermo Fisher Scientific, Waltham, MA, USA).

cDNA synthesis was done by mixing 2 μg of total RNA with 2 μL of oligo (dT) at 2 μg/mL (Promega), and RNAse-free water up to a total volume of 15 μL. Samples were incubated at 70 °C for 5 min in a thermal cycler (C1000 Touch, Bio-Rad Laboratories, Hercules, CA, USA). Then, samples were cooled in ice. Afterward, samples were mixed with 5 μL of M-MLV 5× reaction buffer (Promega), 1.25 μL of 10 mM dNTP mix (Promega), 7 μL of RNAse inhibitors (Promega), 1 μL of M-MLV RT (Promega), and RNAse-free water up to a total volume of 25 μL. Finally, samples were incubated at 37 °C for 60 min.

mRNA expression was assessed by real-time PCR. Briefly, 1 μL of cDNA was mixed with 10 μL of SybrGreen master mix, 1 μL of each primer (10 μM), and 3 μL of RNAse-free water. Then, samples were incubated under the following conditions: pre-incubation, 95 °C for 10 min; denaturation, 40 cycles at 95 °C for 10 s; primer alignment, 56 °C for 10 s; elongation, 72 °C for 10 s. Melting curves were acquired with the following gradient: 95 °C for 10 s; 65 °C for 60 s; and 97 °C for 1 s. Amplification was assessed for the following transcripts *Fasn, Acaca, Acadm, Cpt1, Cd36,* and *Fatp5* using the primers and annealing temperature described in the Appendix A. mRNA relative expression was calculated by normalization against *Actin* according to the 2^−ΔΔCt^ method [14].

#### 2.4.6. Liver Histology Analysis

Formalin-fixed livers were cleared with xylene, then hydrated with gradient ethanol solutions. Then, samples were embedded in paraffin at 60 °C and sectioned at 5 μm. Finally, samples were stained with Hematoxylin and Eosin (H&E) solution, dewaxed, and dehydrated with gradient ethanol solutions. Samples were observed and photographed under a microscope at 40×, analyzing six sections per animal [15].

#### 2.4.7. Extraction and Identification of Urinary Polyphenol Metabolites

Urine samples were subjected to solid phase extraction (SPE) to analyze polyphenol-derived colonic and phase II metabolites. Supel-Select HLB SPE (60 mg/3 mL) cartridges were activated with 3 mL of methanol and 3 mL of water. Then, 2 mL of urine samples were added, followed by a clean-up with 2 mL of 1.5 M formic acid and 2 mL of water:methanol 95:5 (*v*/*v*) to remove interferents. Finally, polyphenol-derived metabolites were eluted with 2 mL of methanol [16].

UPLC-Q-ToF MS was used to assess the polyphenol-derived metabolites profile by following the methodology described in Section 2.3. An aliquot of 1 mL of the samples eluted by SPE was evaporated to dryness (Speedvac Savant, Thermo Fisher Scientific) and resuspended in 200 μL of methanol. For quantification, calibration curves were constructed with 4-hydroxyphenyl propionic acid (phenyl propionic acids and valerolactones), 4-hydroxyphenyl acetic acid (phenylacetic acids), enterodiol (lignans), and hippuric acid (glycinate benzoic acids).

### 2.5. Statistical Analysis

All data are described as mean values ± standard deviation. For the berry-based characterization, three experimental replicates were carried out, and three technical replicates were analyzed. In contrast, ten biological replicates were used for the in vivo experimental design, and three technical replicates were analyzed in each assay. Outliers (>1.5 QR) and extreme (>3.0 QR) values were identified using box-and-whisker plots, and extreme values were excluded. Kolmogorov–Smirnov’s test was used for normality evaluation and Levene’s test for variance homogeneity assessment. Then, multiple comparisons were carried out with Tukey’s test for parametric variables and Kruskal–Wallis for non-parametric parameters. Differences with *p* < 0.05 were considered significant. All statistical analyses were carried out in JMP software v16. Principal Component Analysis (PCA), sparse Partial Least Square-Discriminant Analysis (sPLS-DA), and K-means plots were constructed with the urinary metabolites profile of each experimental group using the Metaboanalyst 5.0 online software. The preprocessing steps included data normalization by sum, square root transformation, and auto-scaling.

## 3. Results

### 3.1. Polyphenol Characterization of the Strawberry, Blueberry, and Strawberry-Blueberry Blend Beverages

The polyphenol composition of the berry fruit-based beverages is shown in Table 1. Sixteen non-pigmented flavonoids, ten anthocyanins, twenty-five phenolic acids, and four ellagitannins were identified in the berry fruit-based beverages. Regarding non-pigmented flavonoids, the blueberry beverage and strawberry-blueberry beverage showed high concentrations of quercetin rhamnoside and quercetin hexoside, which were not identified in the strawberry beverage. Conversely, the strawberry beverage showed a high content of procyanidin dimer B2, eriodictyol, kaempferol hexoside, and (+)-catechin.

Nevertheless, the most significant flavonoids identified in all the berry-based beverages were anthocyanins. The strawberry beverage showed pelargonidin hexoside as the significant anthocyanin, followed by pelargonidin rutinoside, which was not identified in the blueberry beverage. Interestingly, the blueberry beverage showed a richer profile due to its wide variety of anthocyanins. This beverage showed malvidin hexoside as the significant component, followed by malvidin pentoside, peonidin hexoside, cyanidin hexoside, petunidin hexoside, and delphinidin hexoside. As expected, the strawberry-blueberry blend beverage showed a combination of the anthocyanin profile of the individual strawberry and blueberry beverages.

Regarding phenolic acids, the strawberry and strawberry-blueberry blend beverages showed high concentration of hydroxybenzoic acid hexoside and coumaric acid hexoside. In contrast, the blueberry beverage showed a poor profile of hydroxybenzoic acids but showed a high content of chlorogenic acid (caffeoylquinic acid isomer II). In addition, ellagic acid and ellagitannins, mainly peduncalagin and strictinin, were identified in the strawberry and strawberry-blueberry blend beverages, which were not detected in the blueberry beverage.

### 3.2. Effect of the Strawberry, Blueberry, and Strawberry-Blueberry Blend Beverages on Obesity in High-Fat and High-Fructose Diet-Fed Rats

HFF diet-fed rats were supplemented with berry-based beverages for 18 weeks to evaluate their effect on obesity-related metabolic alterations. The effect of berry-based beverages on the biometric and adipose tissue measurements in HFFD-fed rats is shown in Table 2. As expected, after 18 weeks of induction, the HFF diet-fed (HFFD) control group showed increased body weight gain as compared to the standard diet-fed (SD) control group (1.23-fold, *p* = 0.0002), leading to an augmented BMI (1.21-fold, *p* = 0.0006), thus indicating the development of diet-induced obesity.

Moreover, the HFFD control group showed increased triglyceride accumulation in adipose tissue (1.31-fold, *p* = 0.0026), and an increased mesenteric (2.83-fold, *p* = 0.0007), epididymal (2.17-fold, *p* < 0.0001), perirenal (2.64-fold, *p* < 0.0001), and total adipose tissue relative weight (2.48-fold, *p* < 0.0001), leading to an augmented adiposity index (2.00-fold, *p* < 0.0001) as compared to the SD control group. Rats supplemented with berry-based beverages showed a similar daily diet and beverage intake as compared to the HFFD control group (22.3–22.0 vs. 21.5 g/day/rat and 40.7–42.1 vs. 41.0 mL/day/rat, respectively). Nevertheless, none of the berry-based beverages significantly decreased obesity-related parameters compared to the HFFD control group.

### 3.3. Effect of the Strawberry, Blueberry, and Strawberry-Blueberry Blend Beverages on Insulin Resistance in High-Fat and Fructose Diet-Fed Rats

The effect of berry-based beverages on insulin resistance parameters in HFFD-fed rats is shown in Table 2. The HFFD control group showed similar serum glucose levels as compared to the SD control group. Nevertheless, the HFFD control group showed increased serum insulin values (2.03-fold, *p* < 0.0001), leading to the development of insulin resistance, as observed in high HOMA-IR and TyG index and low FGIR index values, and low pancreatic β-cell function, as observed in high HOMA-Beta index values as compared to the SD control group. Regarding the supplementation with berry-based beverages, no significant differences were observed in all the insulin resistance parameters compared to the HFFD control group.

### 3.4. Effect of the Strawberry, Blueberry, and Strawberry-Blueberry Blend Beverages on Triglyceride Metabolism in High-Fat and Fructose Diet-Fed Rats

The effect of berry-based beverages on serum, fecal, and hepatic triglycerides in HFFD-fed rats is shown in Table 3. The HFFD control group showed increased serum and hepatic triglyceride levels (2.45- and 2.63-fold, respectively; *p* = 0.0008 and *p* < 0.0001) compared to the SD control group. Moreover, the administration of the HFFD increased the fecal triglyceride excretion compared to the SD control group (1.43-fold, *p* = 0.0400). Interestingly, the three berry-based beverages decreased serum triglyceride levels compared to the HFFD control group (1.29–1.78-fold, *p* = 0.0048 and *p* = 0.0139). Similarly, all berry-based beverages reduced the accumulation of hepatic triglycerides in comparison with the HFFD control group (1.38–1.63-fold, *p* = 0.0009, *p* = 0.0012, and *p* = 0.0233). Moreover, the strawberry and blueberry beverage supplemented HFFD groups showed similar serum and hepatic triglyceride values as compared to the SD control group. On the other hand, the strawberry beverage slightly increased the fecal triglyceride excretion compared to the HFFD control group (1.45-fold, *p* = 0.0067); however, no significant differences were observed.

Figure 1 shows the liver histology analysis of the experimental groups. The HFFD control group developed hepatic steatosis observed in the presence of lipid vacuoles within the hepatocytes (Figure 1B), whereas the SD control group showed no lipid vacuoles (Figure 1A). Interestingly, the three berry-based beverages decreased the accumulation of lipid vacuoles (Figure 1C,D,E) compared to the HFFD control group (Figure 1B).

Regarding the hepatic lipid metabolism, Figure 2 shows the effect of berry-based beverages on crucial genes involved in fatty acid de novo synthesis, intracellular transport, and β-oxidation. The HFFD control group showed an increased expression of *Acaca* and *Fasn* (1.40- and 1.49-fold, respectively, *p* = 0.0022 and *p* < 0.0001, Figure 2A) and *Fatp5* and *Cd36* (1.53-fold, *p* = 0.0017 and *p* = 0.0002, Figure 2B), and a decreased expression of *Cpt1* and *Acadm* (1.69- and 1.67-fold, respectively, *p* = 0.0451 and *p* = 0.0201, Figure 2C) as compared to the SD control group.

Interestingly, the three berry-based beverages down-regulated *Fasn* expression as compared to the HFFD control group (1.39–1.62-fold, *p* < 0.0001, *p* = 0.0002 and *p* = 0.0018), showing similar expression values to the SD control group, whereas the blueberry and strawberry-blueberry blend beverages down-regulated *Acaca* expression as compared to the HFFD control group (1.93- and 1.59-fold, respectively, *p* < 0.0001 and *p* = 0.0003).

The effect of berry-based beverages on the hepatic fatty acid profile in HFFD-fed rats is shown in Table 3. As expected, the HFFD control group showed a significant (*p* < 0.05) increased concentration of most hepatic fatty acids as compared to the SD control group, except for EPA (*cis*-5,8,11,14,17-eicosapentaenoic acid, and c205ω3), which was slightly decreased (1.50-fold). Interestingly, the strawberry and blueberry-based beverages slightly decreased the accumulation of saturated (1.37- and 1.25-fold, respectively), monounsaturated (1.36- and 1.30-fold, respectively), and polyunsaturated (1.35- and 1.15-fold, respectively) fatty acids as compared to the HFFD control group; however, no significant differences were observed. Even though the three berry-based beverages showed a similar beneficial effect on hepatic triglyceride accumulation, the strawberry beverage showed a more significant effect on the hepatic fatty acid profile.

### 3.5. Urinary Polyphenol Metabolites Associated with the Supplementation of Strawberry, Blueberry, and Strawberry-Blueberry Blend Beverages in High-Fat and High-Fructose Diet-Fed Rats

The effect of berry-based beverages on the urinary polyphenol metabolite profile in HFFD-fed rats is shown in Appendix A. Twenty-three polyphenol metabolites were identified in SD- and HFFD-fed rats, mainly flavone and isoflavone metabolites, such as apigenin glucuronide, equol glucuronide, and hydroxydaidzein. Interestingly, the strawberry beverage excreted the greatest variety of polyphenol metabolites (n = 48), with urolithins A and B as significant compounds, which were not detected in the control groups. The blueberry and strawberry-blueberry blend beverages showed a lower variety of urinary metabolites (n = 33 and 30, respectively), where urinary enterolactone was significantly increased compared to the HFFD control group (3.02- and 2.55-fold, respectively, *p* = 0.008). Multivariate analyses of the urinary polyphenol metabolite profile are shown in Figure 3, Figure 4 and Figure 5.

The unsupervised PCA and the supervised sPLS-DA models showed a total explained variance of 54.8% and 54.5%, respectively, showing the discrimination between some experimental groups (Figure 3A and Figure 4A, respectively). The greatest variance was explained by component 1 (x-axis), which discriminated between rats fed a SD, a HFFD, and a HFFD supplemented with the blueberry and blueberry-strawberry beverages and those rats fed a HFFD supplemented with the strawberry beverage. Similar results were observed in the K-means clustering analysis, which also confirmed the integration of three clear clusters: cluster 1: rats fed a HFFD supplemented with the blueberry and blueberry-strawberry beverages; cluster 2: rats fed a HFFD supplemented with the strawberry beverage; and cluster 3: rats fed with SD and HFFD (control groups), indicating that the urinary metabolite profile is similar between those rats included in each cluster.

The urinary metabolites responsible for the discrimination between the experimental groups are observed in the biplot (Figure 3B), VIP score plots (Figure 4C,D), and K-means features (Figure 5B). Interestingly, several polyphenol urinary metabolites were identified as descriptors of the rats fed a HFFD supplemented with the strawberry beverage, such as catechol sulfate, urolithin B glucuronide, urolithin C, dihydroferulic acid, methylurolithin A, methylpyrogallol sulfate, and hydroxyhippuric acid (loading score > 0.25; Appendix A); whereas the primary discriminant metabolites of the rats fed a HFFD supplemented with the blueberry and strawberry-blueberry beverages were methyl-(epi)-catechin glucuronide, dihydroxyphenylvalerolactone, and enterolactone (loading score > 0.25; Appendix A). On the other hand, both control groups showed methoxyhydroxyphenylvalerolactone, equol glucuronide, dimethyl quercetin, and hydroxyglicitein as discriminant urinary metabolites (loading score > 0.25; Appendix A).

## 4. Discussion

Numerous experimental studies (in vitro, in vivo, and clinical trials), and a meta-analysis of epidemiological studies, have demonstrated that berry fruits can prevent or counteract diet-induced obesity and obesity-related complications due to their high content of bioactive compounds, mainly dietary fiber, vitamins, and polyphenols [17]. Therefore, in this study, we hypothesized that functional beverages could be developed with decoctions elaborated with polyphenol-rich berry fruits, such as strawberries and blueberries, which can exert some of the health beneficial effects associated with the consumption of whole-berry fruits since polyphenols are extracted during the decoction process used in the elaboration of functional beverages as previously demonstrated [4].

All the berry-based beverages developed in this study showed a high content of polyphenols, mainly anthocyanins. As expected, contrasting profiles were identified in strawberry and blueberry-based beverages since the strawberry-based beverage showed pelargonidin hexoside as a the main polyphenol, followed by several hydroxycinnamic acids and ellagitannins, such as coumaric acid hexoside, ellagic acid, peduncalagin, and strictinin, while the blueberry-based beverage was rich in several flavonoids, including malvidin hexoside, malvidin pentoside, peonidin hexoside, quercetin rhamnoside, quercetin hexoside, and in chlorogenic acid. These primary polyphenols agree with those identified in both strawberry and blueberry fruits [6,18].

In this study, we attempt to demonstrate the bioactivity of the berry fruits beverages using an in vivo model induced with a HFFD for eighteen weeks that led to the development of obesity accompanied by insulin resistance, impaired function of β-pancreatic cells, hypertriglyceridemia, and hepatic steatosis. The supplementation with all berry-based beverages did not prevent body weight gain nor the accumulation of triglycerides in adipose tissue induced by the HFFD. In this regard, Prior et al. [19] demonstrated that strawberry and blueberry fruits did not prevent weight gain in high-fat diet-fed rats. In contrast, an equivalent concentration of purified anthocyanins of these berry fruits exerted antiobesogenic effects. Moreover, a methanolic strawberry extract promoted adipocyte browning and inhibited adipogenesis in 3T3 L1 cells [18], whereas blueberry ethanolic extract increased energy expenditure in brown adipose tissue and adipocyte browning in the inguinal white adipose tissue [20].

Similarly, the berry-based beverages developed in this study did not exert a protective effect against developing HFFD-induced hyperglycemia, hyperinsulinemia, insulin resistance, and β-cell pancreatic damage. Conversely, Liu et al. [21] reported that the supplementation with freeze-dried whole blueberry powder increased insulin sensitivity and glucose tolerance in high-fat diet-induced mice, improving b-cell survival, and preventing β-cell hypertrophy. In contrast, Aranaz et al. [7] demonstrated that a freeze-dried strawberry and blueberry blend powder reduced hyperinsulinemia and insulin resistance in high-fat and high-sucrose diet-fed obese rats. Notably, the polyphenol content of the berry extracts or powders of these previous studies was higher than those found in our study, which could be partly related to the lack of an anti-obesogenic and insulin sensitizer effect of our berry-based beverages.

On the other hand, the three berry-based beverages developed in our study exerted a chronic anti-hypertriglyceridemic effect as observed in decreased serum triglyceride levels as compared to the HFFD control group, which was not associated with an increased fecal triglyceride excretion, but was accompanied by the prevention of the development of the fatty liver. Furthermore, all berry-based beverages decreased the hepatic accumulation of triglycerides, whereas the strawberry beverage decreased the hepatic accumulation of saturated fatty acids. Accordingly, Wang et al. [22] reported that anthocyanin-rich extracts of wild blueberry and strawberry fruits decreased triglyceride accumulation in HepG2 cells induced with oleic acid to simulate an in vitro fatty liver. Interestingly, when purified anthocyanins were evaluated, cyanidin 3-O-glucoside and delphinidin 3-O-glucoside exerted the greatest clearance of hepatic triglycerides, anthocyanins identified in both beverages elaborated with blueberries, but not in the 100% strawberry-based beverage. Moreover, Liu et al. [23] demonstrated that the blueberry phenolic acid-rich fraction exerted a more significant inhibitory effect on triglyceride accumulation as compared to the anthocyanin-rich fraction in oleic acid-induced HepG2 cells, which was partly related to the improvement of triglycerides clearance by caffeic acid and chlorogenic acid, this latter was the major phenolic acid identified in the blueberry-based beverage.

The hepatic accumulation of triglycerides in a combination with obesity and insulin resistance is known as metabolic-associated fatty liver disease (MAFLD), which is associated with an imbalance of hepatic lipid metabolism pathways, such as the import/export of fatty acids, de novo fatty acid biosynthesis (lipogenesis), and fatty acid catabolism (β-oxidation) [24]. The HFFD model used for this study developed a mild stage of MAFLD as observed in the accumulation of lipid vacuoles within hepatocytes without the elevation of hepatic injury markers (ALT, AST, and ALP). Moreover, the HFFD control group showed an increased expression of *Acaca* and *Fasn* (lipogenesis), and *Fatp5* and *Cd36* (fatty acid uptake), and a decreased expression of *Cpt1* and *Acadm* (β-oxidation).

Hepatic de novo lipogenesis is mainly regulated by acetyl-CoA carboxylase (ACC), which converts acetyl-CoA to malonyl-CoA. Then, FAS carries out the conversion to palmitate, which is further elongated and desaturated for the synthesis of multiple fatty acids, which can be further esterified for the synthesis of triglycerides. Interestingly, all berry-based beverages down-regulated *Fasn* expression, whereas both beverages elaborated with blueberries down-regulated *Acaca* expression, achieving a similar expression compared to the SD control group. However, only the supplementation with the strawberry-based beverage significantly reduced the hepatic accumulation of palmitic acid, a well-known lipotoxic compound that promotes the synthesis of ceramides and diglycerides that further promote the transition from MAFLD to the proinflammatory non-alcoholic steatohepatitis (NASH) [24].

Fatty acid β-oxidation is mainly regulated by the CPT1 enzyme, which is responsible for the transport of fatty acids into the mitochondria. In contrast, acyl-CoA dehydrogenases (ACAD) participate in the first step of mitochondrial fatty acid oxidation. All berry-based beverages up-regulated *Cpt1a* and *Acadm* hepatic expression; however, the strawberry-based beverage showed higher expression values than the SD control group. In addition, hepatocytes control the flux of fatty acids via fatty acid transport proteins (FATP5) and CD36, which are commonly increased in high-fat diet-induced MAFLD [24]. The supplementation with the 100% strawberry- and 100% blueberry-based beverages down-regulated *Cd36* expression, whereas the blueberry- and the strawberry-blueberry blend-based beverages down-regulated *Fatp5* expression.

Interestingly, even though different genes were up- or down-regulated by each berry-based beverage, the three beverages exerted a similar reduction effect on the accumulation of lipid vacuoles within the hepatocytes. Similar results were reported with the supplementation of wild blueberry fruits, which up-regulated the expression of the peroxisome proliferator-activated receptor-alpha (PPAR-α) in obese rats, which regulates the expression of *Cpt1a* and down-regulated the expression of the sterol regulatory element-binding protein 1 (SREBP-1), which regulates the expression of *Fasn* and *Acaca* genes [25]. On the other hand, the effect of strawberry fruit or derived products has not been reported on hepatic lipid metabolism. Nevertheless, Zhang et al. [26] demonstrated that ellagic acid, which was only detected in strawberry-based beverages, attenuates the development of hepatic steatosis by the inhibition of the activity and transcription of hepatic SREBP-1, FAS, and ACC. Moreover, punicalagin, the major ellagitannin identified in the strawberry-based beverage, and ellagic acid exert a protective effect against palmitate-induced mitochondrial dysfunction in HepG2 cells [27].

Finally, in this study, we conducted a targeted metabolomic analysis of urine samples obtained to identify the polyphenol metabolites associated with the chronic daily supplementation of berry-based beverages. Notably, several polyphenol urinary metabolites were identified in all the experimental groups, which is associated with the polyphenol intake of rodent diet that is elaborated with soybean as one of the major ingredients. All animals excreted a relatively high amount of apigenin glucuronide, equol glucuronide, and hydroxydaidzein. Accordingly, soybeans are uniquely rich in isoflavones, such as daidzin, genistin, and glycitin, which are hydrolyzed into their aglycones (daidzein, genistein, and glycitein, respectively) in the small intestine, and could be further absorbed intact or metabolized by colonic microbiota, which catalyzes their hydroxylation or glucuronidation and can promote the formation of equol (7,4′-isoflavandiol) [28,29]. Moreover, both control groups (SD and HFFD) showed a similar urinary polyphenol metabolite profile according to the chemometric analysis, since their polyphenol intake was exclusively provided by the standard rodent diet. The main metabolites identified as discriminants of the control groups were equol glucuronide (major urinary metabolite) and several minor components, such as methoxyhydroxyphenylvalerolactone, dimethyl quercetin, and hydroxyglycitein.

According to the multivariate analysis, the main urinary metabolites associated with the consumption of the strawberry-based beverage included catechol sulfate, dihydroferulic acid, methylpyrogallol sulfate, and hydroxyhippuric acid, and urolithin B glucuronide, urolithin C, and mehtylurolihin A. Truchado et al. [30] identified urolithin A glucuronide as the predominant urinary metabolite after the consumption of whole strawberry fruits, followed by urolithin A, urolithin B, and urolithin B glucuronide, which are produced during the colonic fermentation of ellagitannins and ellagic acid. It is noteworthy that ellagitannins are partially hydrolyzed into ellagic acid in the small intestine. However, these polyphenols have a low bioavailability; therefore, most of them reach the colon, where are subjected to gut microbiota metabolism, including the formation of urolithins and their further sulfatation, methylation, and glucuronidation [31]. Interestingly, the formation of urolithins after strawberry intake is not affected by the food processing process [31]; nevertheless, ellagitannins are not exclusive to strawberry since these polymeric polyphenols are also found in pomegranate and several nuts, and even though each source has a unique ellagitannin profile, the urinary metabolite profile is similar [32].

On the other hand, previous studies have identified anthocyanin metabolites after consuming different berry fruits and derived products [32]; however, these metabolites were not identified in our study. Nevertheless, catechol sulfate and hydroxyhippuric acid were identified as discriminant urinary metabolites of the strawberry-based beverage, which are metabolites derived from the colonic fermentation of several anthocyanins [33].

Interestingly, even though the strawberry-blueberry blend-based beverage showed a mixed polyphenol profile, its urinary polyphenol metabolite profile was similar to that found for the 100% blueberry-based beverage, characterized by the excretion of methyl-(epi)-catechin glucuronide, dihydrophenylvalerolactone, and enterolactone. Accordingly, Ancillotti et al. [34] and Langer et al. [35] also identified phenylvalerolactone derivatives as blueberry fruit and juice urinary derivatives, respectively. However, these metabolites cannot be considered exclusive from blueberry consumption since valerolactones are produced during the colonic fermentation of flavanals and proanthocyanidins (polymeric flavanals), which are extensively distributed in fruits, vegetables, nuts, and seeds [36]. Interestingly, enterolactone is a colonic microbiota derivative from lignans [37]; however, lignans were not identified in the blueberry-based beverage. However, secoisolariciresinol and matairesinol are the main lignans identified in berry fruits [38]. Therefore, we hypothesized that lignans were lower than the detection limit in our berry-based beverages. However, their colonic metabolites could have been accumulated in the urinary bladder and thus were detected as urinary polyphenol metabolites. Even though the discriminant urinary polyphenol metabolites identified in this study are not exclusive of strawberry and blueberry consumption, they can be associated with the health-beneficial effects exerted by berry-based beverages. For instance, urolithin A and urolithin B down-regulate *Srepb1a* and up-regulate *Ppara* hepatic expression in high-fat diet-fed rats, decreasing the accumulation of hepatic triglycerides [39].

Finally, it is noteworthy that the main health-beneficial effect exerted by the berry fruit decoction-based beverages was the prevention of hypertriglyceridemia and hepatic steatosis. Even though we demonstrated that strawberry and blueberry blended beverage showed a greater diversity of polyphenols, this beverage exerted a slightly lower preventive effect on these metabolic alterations as compared to the single fruit-based beverage, which could be related to the complex synergetic, additive, and antagonistic interactions between polyphenols that affect their global health impact [40].

## 5. Conclusions

The results obtained in this study demonstrate that berry fruits, such as strawberries and blueberries, can be used to develop functional beverages to prevent high-fat and high-fructose diet-induced hypertriglyceridemia and hepatic steatosis through the modulation of key genes involved in fatty acid hepatic metabolism. Furthermore, these health-beneficial effects found in berry fruit decoction-based beverages were associated with their polyphenol composition, including urinary metabolites, such as urolithins. Nevertheless, further studies must be carried out to confirm the results obtained in this in vivo study through placebo-controlled clinical trials and understand the synergetic, additive, or antagonistic interactions between polyphenols.

## Figures and Tables

**Figure 1 ijerph-20-04418-f001:**
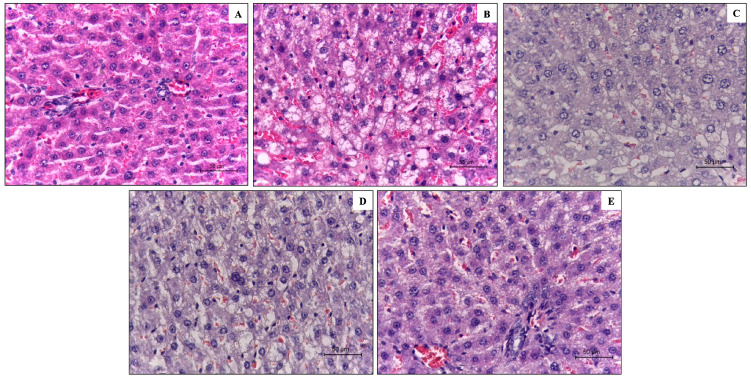
Liver histological analysis of standard diet diet-fed rats (**A**), high-fat and high-fructose diet-fed rats (**B**), high-fat and high-fructose diet-fed rats treated with strawberry beverage (**C**), high-fat and high-fructose diet-fed rats treated with blueberry beverage (**D**), and high-fat and high-fructose diet-fed rats treated with strawberry-blueberry blend beverage (**E**). Magnification at 40×.

**Figure 2 ijerph-20-04418-f002:**
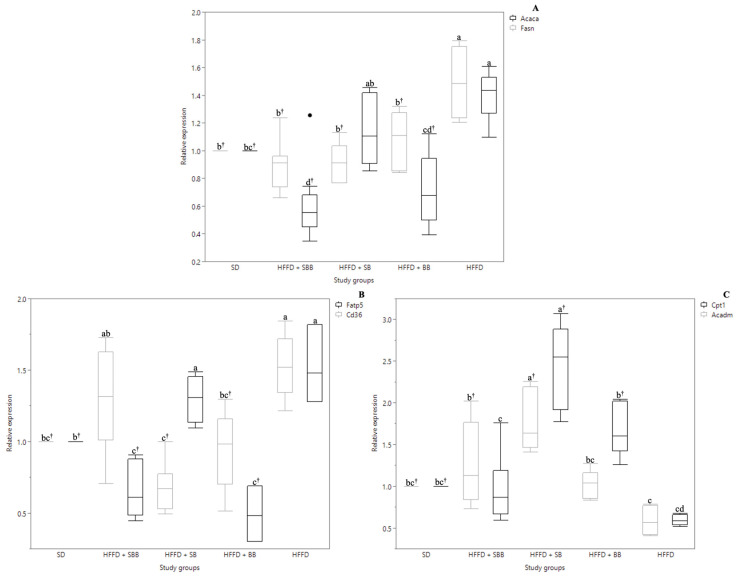
Effect of the strawberry, blueberry, and strawberry-blueberry blend beverages on genes involved in fatty acid de novo synthesis (**A**), fatty acids transport (**B**), and fatty acids β-oxidation (**C**) in high-fat and high-fructose diet-fed rats. Data are shown as mean values, and error bars represent the interquartile range (n = 10). Different letters indicate significant (*p* < 0.05) differences between samples by Tukey’s or Kruskal–Wallis’s test. ^†^ Indicate significant (*p* < 0.05) difference compared to the HFFD group by Dunnett’s or Wilcoxon test. SD: standard diet; HFFD: high-fat and high-fructose diet; SB: strawberry beverage; BB: blueberry beverage; SBB: strawberry-blueberry beverage.

**Figure 3 ijerph-20-04418-f003:**
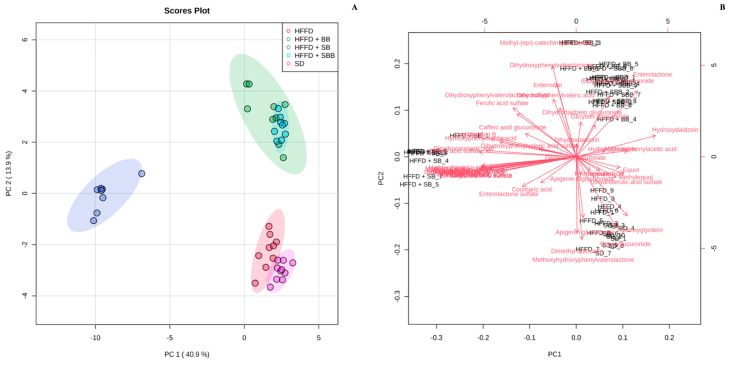
Principal Component Analysis plot (**A**) and biplot (**B**) of urinary polyphenol metabolites of high-fat and high-fructose diet-fed rats supplemented with strawberry, blueberry, and strawberry-blueberry blend beverages. Data were normalized by sum, square root transformed, and auto-scaled. SD: standard diet; HFFD: high-fat and high-fructose diet; SB: strawberry beverage; BB: blueberry beverage; SBB: strawberry-blueberry beverage.

**Figure 4 ijerph-20-04418-f004:**
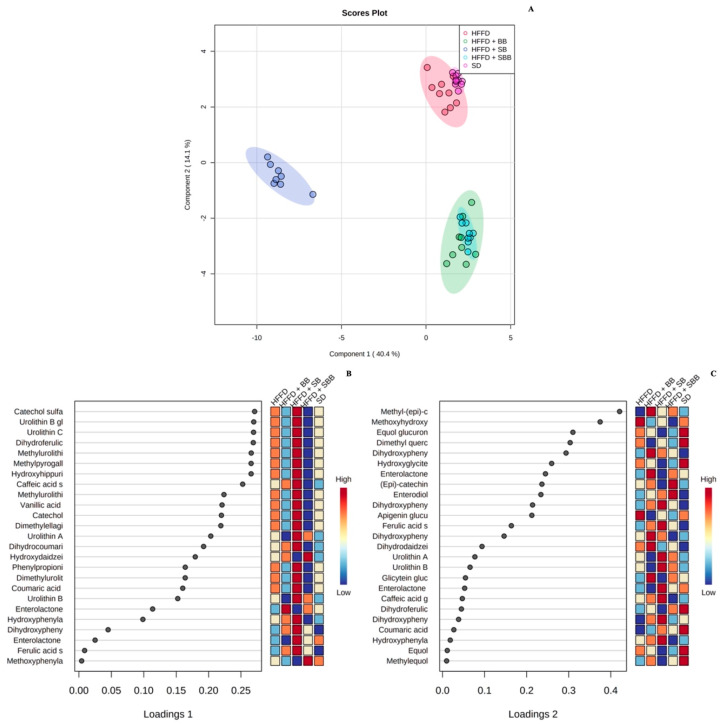
Spare Partial Least Square-Discriminant Analysis plot (**A**), loadings of component 1 (**B**), and loadings of component 2, (**C**) of urinary polyphenol metabolites of high-fat and high-fructose diet-fed rats supplemented with strawberry, blueberry, and strawberry-blueberry blend beverages. Data were normalized by sum, square root transformed, and auto-scaled. SD: standard diet; HFFD: high-fat and high-fructose diet; SB: strawberry beverage; BB: blueberry beverage; SBB: strawberry-blueberry beverage.

**Figure 5 ijerph-20-04418-f005:**
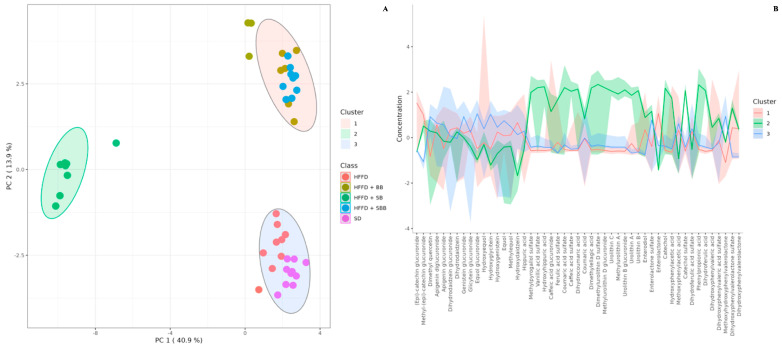
K-means clustering plot (**A**) and features (**B**) of urinary polyphenol metabolites of high-fat and high-fructose diet-fed rats supplemented with strawberry, blueberry, and strawberry-blueberry blend beverages. Data were normalized by sum, square root transformed, and auto-scaled. SD: standard diet; HFFD: high-fat and high-fructose diet; SB: strawberry beverage; BB: blueberry beverage; SBB: strawberry-blueberry beverage.

**Table 1 ijerph-20-04418-t001:** Polyphenol composition and antioxidant capacity of the strawberry (SB), blueberry (BB), and strawberry-blueberry blend beverages (SBB).

Tentative Identification	Rt (min)	Molecular Formula	Expected Mass (Da)	Observed Mass (Da)	Mass Error (ppm)	Concentration (μg/mL)		
						SB	BB	SBB
Polyphenol composition ^1^								
Flavanols								
(+)-Catechin hexoside	2.08	C_21_H_24_O_11_	452.1319	451.1239	−1.4100	0.44 ± 0.05 ^a^	0.03 ± 0.00 ^c^	0.21 ± 0.01 ^b^
Procyanidin dimer B2	2.48	C_30_H_26_O_12_	578.1424	577.1338	−2.2829	3.03 ± 0.46 ^a^	0.94 ± 0.05 ^c^	1.79 ± 0.04 ^b^
(−)-Epicatechin hexoside	2.60	C_21_H_24_O_11_	452.1319	451.1238	−1.8072	0.31 ± 0.00 ^a^	ND	0.16 ± 0.02 ^b^
(+)-Catechin *	2.68	C_15_H_14_O_6_	290.0790	289.0713	−1.7489	1.91 ± 0.07 ^a^	0.34 ± 0.01 ^c^	0.99 ± 0.06 ^b^
Procyanidin trimer C1	2.69	C_45_H_38_O_18_	866.2058	865.1969	−1.8700	0.96 ± 0.30 ^a^	0.02 ± 0.01 ^b^	0.16 ± 0.03 ^b^
(−)-Epicatechin *	3.12	C_15_H_14_O_6_	290.0790	289.0717	−0.3075	0.06 ± 0.01 ^c^	0.27 ± 0.01 ^a^	0.16 ± 0.01 ^b^
Flavanones								
Eriocitrin	2.93	C_27_H_32_O_15_	596.1741	595.1659	−1.6436	0.54 ± 0.00 ^a^	0.05 ± 0.01 ^c^	0.31 ± 0.02 ^b^
Naringenin hexoside	4.18	C_21_H_22_O_10_	434.1213	433.1150	2.3310	0.04 ± 0.00 ^c^	0.08 ± 0.00 ^a^	0.06 ± 0.00 ^b^
Eriodictyol	4.65	C_15_H_12_O_6_	288.0634	287.0561	0.0945	2.27 ± 0.82 ^a^	ND	0.23 ± 0.02 ^b^
Flavonols								
Myricetin hexoside	3.48	C_21_H_20_O_13_	480.0904	479.0826	−1.0119	ND	0.44 ± 0.02 ^a^	0.24 ± 0.05 ^b^
Quercetin hexoside	4.24	C_21_H_20_O_12_	464.0955	463.0890	1.7997	ND	6.47 ± 1.13 ^a^	3.15 ± 0.22 ^b^
Kaempferol xylosyl-hexoside	4.43	C_26_H_28_O_15_	580.1428	579.1356	0.0895	0.09 ± 0.01 ^a^	ND	0.05 ± 0.01 ^a^
Kaempferol hexoside-rhamnoside	5.12	C_27_H_30_O_15_	594.1585	593.1509	−0.5466	0.73 ± 0.09 ^a^	ND	0.28 ± 0.02 ^b^
Kaempferol hexoside	5.44	C_21_H_20_O_11_	448.1006	447.0940	1.5585	2.07 ± 0.19	ND	ND
Quercetin rhamnoside	5.52	C_21_H_20_O_11_	448.1006	447.0940	1.6864	ND	7.56 ± 0.40 ^a^	5.84 ± 0.47 ^b^
Kaempferol acetyl-hexoside	6.52	C_23_H_22_O_12_	490.1111	489.1044	1.2081	0.20 ± 0.04 ^a^	ND	0.07 ± 0.00 ^b^
Anthocyanins								
Pelargonidin rutinoside-hexoside	2.08	C_33_H_41_O_19_	741.2262	741.2257	2.7129	0.01 ± 0.00	ND	ND
Delphinidin hexoside	2.27	C_21_H_21_O_12_	465.1031	465.1025	−0.4505	ND	2.23 ± 0.30 ^a^	1.16 ± 0.15 ^b^
Cyanidin hexoside	2.51	C_21_H_21_O_11_	449.1084	449.1078	−0.0460	ND	5.41 ± 0.27 ^a^	2.44 ± 0.22 ^b^
Petunidin hexoside	2.63	C_22_H_23_O_12_	479.1187	479.1181	−0.6125	ND	3.74 ± 0.28 ^a^	1.62 ± 0.11 ^b^
Pelargonidin hexoside	2.82	C_21_H_21_O_10_	433.1121	433.1116	−3.1325	35.80 ± 0.07 ^a^	ND	16.02 ± 0.28 ^b^
Malvidin pentoside	2.89	C_22_H_23_O_11_	463.1238	463.1232	−0.5449	ND	8.32 ± 0.51 ^a^	4.42 ± 0.45 ^b^
Pelargonidin rutinoside	2.90	C_27_H_31_O_14_	579.1705	579.1699	−1.5811	4.06 ± 0.32 ^a^	ND	1.40 ± 0.03 ^b^
Malvidin hexoside	2.93	C_23_H_25_O_12_	493.1344	493.1338	−0.5038	ND	15.34 ± 0.71 ^a^	8.31 ± 0.80 ^b^
Peonidin hexoside	3.10	C_22_H_23_O_11_	463.1237	463.1232	−0.6543	ND	8.01 ± 0.50 ^a^	3.69 ± 0.21 ^b^
Pelargonidin succinyl-hexoside	3.47	C_25_H_25_O_13_	533.1314	533.1309	3.5346	ND	0.08 ± 0.01	ND
Hydroxybenzoic acids								
Galloylquinic acid isomer I	1.53	C_14_H_16_O_10_	344.0743	343.0661	−2.8640	0.54 ± 0.04 ^a^	ND	0.42 ± 0.01 ^a^
Gallic acid hexoside	1.80	C_13_H_16_O_10_	332.0743	331.0659	−3.5576	2.08 ± 0.06 ^a^	0.19 ± 0.03 ^c^	0.78 ± 0.08 ^b^
Galloylquinic acid isomer II	2.04	C_14_H_16_O_10_	344.0743	343.0662	−2.3997	1.34 ± 0.11 ^a^	0.13 ± 0.02 ^c^	0.60 ± 0.09 ^b^
Hydroxybenzoic acid hexoside	2.08	C_13_H_16_O_8_	300.0845	299.0761	−3.9366	11.00 ± 0.27 ^a^	ND	5.93 ± 0.54 ^b^
Dihydroxybenzoic acid hexoside	2.15	C_13_H_16_O_9_	316.0794	315.0714	−2.2841	1.47 ± 0.00 ^ab^	ND	0.73 ± 0.03 ^b^
Vanillic acid *	2.84	C_8_H_8_O_4_	168.0423	167.0345	−2.8946	ND	0.27 ± 0.00 ^a^	0.15 ± 0.01 ^a^
Syringic acid	3.07	C_9_H_10_O_5_	198.0528	197.0453	−1.4804	ND	0.66 ± 0.05 ^a^	0.27 ± 0.01 ^b^
Dihydroxybenzoic acid isomer I	3.47	C_7_H_6_O_4_	154.0266	153.0189	−2.9487	ND	0.84 ± 0.02	ND
Dihydroxybenzoic acid isomer II	3.62	C_7_H_6_O_4_	154.0266	153.0186	−4.8918	2.65 ± 0.00 ^a^	ND	1.38 ± 0.07 ^b^
Methylgallate	4.72	C_8_H_8_O_5_	184.0372	183.0294	−2.5857	0.09 ± 0.02	ND	ND
Hydroxycinnamic acids								
Caffeoylquinic acid isomer I	2.23	C_16_H_18_O_9_	354.0951	353.0880	0.5868	ND	0.50 ± 0.02 ^a^	0.27 ± 0.02 ^b^
Caffeic acid hexoside	2.38	C_15_H_18_O_9_	342.0951	341.0876	−0.6122	1.71 ± 0.10 ^a^	ND	1.57 ± 0.00 ^a^
Ferulic acid hexoside	2.50	C_16_H_20_O_9_	356.1107	355.1037	0.6177	ND	1.65 ± 0.07 ^a^	0.53 ± 0.10 ^b^
Caffeoylquinic acid isomer II *	2.67	C_16_H_18_O_9_	354.0951	353.0894	4.4918	0.03 ± 0.00 ^c^	14.50 ± 0.45 ^a^	8.00 ± 0.06 ^b^
Sinapic acid hexoside	2.70	C_17_H_22_O_10_	386.1213	385.1149	2.2967	ND	0.38 ± 0.02 ^a^	0.17 ± 0.01 ^b^
Coumaric acid hexoside	2.80	C_15_H_18_O_8_	326.1002	325.0926	−1.0236	24.31 ± 0.72 ^a^	1.27 ± 0.98 ^c^	16.98 ± 0.87 ^b^
Caffeic acid *	2.93	C_9_H_8_O_4_	180.0423	179.0343	−3.5723	0.06 ± 0.01 ^c^	0.38 ± 0.02 ^a^	0.24 ± 0.02 ^b^
Caffeoylquinic acid isomer III	3.09	C_16_H_18_O_9_	354.0951	353.0885	2.0741	ND	0.49 ± 0.04 ^a^	0.26 ± 0.01 ^b^
Ellagic acid hexoside	3.15	C_20_H_16_O_13_	464.0591	463.0518	−0.0246	1.13 ± 0.28 ^a^	ND	0.32 ± 0.04 ^b^
Sinapic acid	3.19	C_11_H_12_O_5_	224.0685	223.0616	1.7576	ND	0.02 ± 0.00	ND
Feruloylquinic acid	3.42	C_17_H_20_O_9_	368.1107	367.1045	2.9639	ND	0.27 ± 0.02 ^a^	0.15 ± 0.01 ^b^
Coumaric acid *	3.60	C_9_H_8_O_3_	164.0473	163.0394	−4.0947	0.41 ± 0.04 ^a^	ND	0.20 ± 0.01 ^b^
Coumaroylquinic acid	3.63	C_16_H_18_O_8_	338.1002	337.0938	2.6326	ND	0.09 ± 0.01 ^a^	0.05 ± 0.01 ^a^
Ellagic acid *	3.94	C_14_H_6_O_8_	302.0063	300.9989	−0.1393	5.73 ± 1.92 ^a^	ND	3.88 ± 0.10 ^b^
Cinnamic acid	4.98	C_9_H_8_O_2_	148.0524	147.0446	−3.9981	0.20 ± 0.03 ^a^	ND	0.07 ± 0.01 ^b^
Ellagitannins								
Peduncalagin	1.92	C_34_H_24_O_22_	784.0759	783.0652	−4.4377	8.07 ± 3.59 ^a^	ND	3.48 ± 0.14 ^b^
Castalin	2.52	C_27_H_20_O_18_	632.0650	631.0562	−2.3169	0.85 ± 0.33 ^a^	ND	0.22 ± 0.05 ^b^
Strictinin	2.70	C_27_H_22_O_18_	634.0806	633.0717	−2.5112	4.25 ± 1.26 ^a^	ND	2.32 ± 0.06 ^b^
Castalagin	3.20	C_41_H_26_O_26_	934.0712	933.0635	−0.5354	0.10 ± 0.04	ND	ND
Antioxidant capacities								
Folin–Ciocalteu reducing capacity assay ^2^						0.97 ± 0.03 ^a^	0.88 ± 0.04 ^b^	0.90 ± 0.02 ^b^
DPPH^+^ radical scavenging assay ^3^						0.90 ± 0.03 ^a^	0.46 ± 0.02 ^c^	0.65 ± 0.04 ^b^
ABTS^-^ radical scavenging assay ^3^						1.15 ± 0.08 ^a^	0.60 ± 0.02 ^c^	0.92 ± 0.04 ^b^

Data are shown as mean ± standard deviation of three replicates. Results are expressed as ^1^ mg/mL, ^2^ mg gallic acid equivalent/mL, and ^3^ mg Trolox equivalent/mL. Different letters indicate significant (*p* < 0.05) differences between samples by Tukey’s test. * Identification confirmed with commercial standards. SB: strawberry beverage; BB: blueberry beverage; SBB: strawberry-blueberry beverage; Rt: retention time; ND: not detected; DPPH: 2,2-diphenyl-1-picrylhydrazyl; ABTS: 2,2′-azino-bis(3-ethylbenzo-thiazoline-6-sulfonic acid); Trolox: 6-hydroxy-2,5,7,8-tetramethylchroman-2-carboxylic acid.

**Table 2 ijerph-20-04418-t002:** Effect of the strawberry, blueberry, and strawberry-blueberry blend beverages on obesity, insulin resistance, and hepatic injury biomarkers in high-fat and high-fructose diet-fed rats.

Parameters	Control Groups		Beverage Treated Groups		
	SD	HFFD	HFFD + SB	HFFD + BB	HFFD + SBB
Biometric measurements					
Body weight (g)	550.6 ± 18.1 ^b†^	675.3 ± 38.6 ^a^	686.3 ± 20.3 ^a^	659.9 ± 28.3 ^a^	672.2 ± 56.4 ^a^
Body mass index (g/cm^2^)	0.76 ± 0.04 ^b†^	0.92 ± 0.05 ^a^	0.90 ± 0.10 ^a^	0.90 ± 0.04 ^a^	0.93 ± 0.03 ^a^
Lee index	0.30 ± 0.01 ^a†^	0.32 ± 0.01 ^a^	0.32 ± 0.02 ^a^	0.32 ± 0.01 ^a^	0.33 ± 0.01 ^a^
Adipose tissue measurements					
Triglycerides (mg/g)	556.7 ± 60.7 ^b†^	730.7 ± 67.1 ^a^	653.4 ± 66.4 ^ab^	611.3 ± 41.7 ^ab^	712.9 ± 79.6 ^a^
Mesenteric relative weight (g/kg)	4.1 ± 1.6 ^b†^	11.5 ± 4.1 ^a^	12.4 ± 2.4 ^a^	10.4 ± 5.8 ^a^	10.9 ± 1.9 ^a^
Epididymal relative weight (g/kg)	10.6 ± 2.2 ^b†^	22.9 ± 7.0 ^a^	25.8 ± 6.4 ^a^	18.0 ± 4.5 ^a^	24.6 ± 6.1 ^a^
Perirenal relative weight (g/kg)	10.6 ± 2.6 ^b†^	28.0 ± 6.6 ^a^	32.5 ± 7.9 ^a^	22.7 ± 10.4 ^a^	30.6 ± 9.2 ^a^
Total adipose tissue relative weight (g/kg)	25.2 ± 5.8 ^b†^	62.4 ± 16.4 ^a^	70.7 ± 14.2 ^a^	51.1 ± 7.5 ^a^	66.2 ± 17.4 ^a^
Adiposity index (%)	4.6 ± 0.6 ^b†^	9.2 ± 1.1 ^a^	10.2 ± 1.0 ^a^	7.9 ± 0.8 ^a^	9.7 ± 1.2 ^a^
Insulin resistance parameters					
Serum glucose (mg/dL)	154.4 ± 317.0 ^a^	160.3 ± 45.8 ^a^	148.2 ± 27.9 ^a^	144.3 ± 22.9 ^a^	151.5 ± 19.9 ^a^
Serum insulin (ng/mL)	3.9 ± 1.3 ^b†^	7.9 ± 2.0 ^a^	8.1 ± 2.0 ^a^	6.6 ± 2.5 ^ab^	7.4 ± 2.9 ^a^
HOMA-IR index	5.8 ± 2.0 ^b†^	15.0 ± 4.3 ^a^	12.9 ± 5.0 ^a^	10.1 ± 5.2 ^a^	11.8 ± 4.9 ^a^
HOMA-Beta index	18.4 ± 8.1 ^b†^	33.6 ± 13.6 ^a^	35.9 ± 9.5 ^a^	30.4 ± 10.9 ^ab^	32.0 ± 12.7 ^ab^
QUICKI	0.24 ± 0.01 ^a^	0.22 ± 0.01 ^a^	0.22 ± 0.01 ^a^	0.23 ± 0.01 ^a^	0.23 ± 0.01 ^a^
FGIR index	1.62 ± 0.53 ^a†^	0.84 ± 0.28 ^b^	0.76 ± 0.14 ^b^	1.00 ± 0.47 ^b^	0.99 ± 0.50 ^b^
TyG index	4.6 ± 0.2 ^a^	5.0 ± 0.2 ^a^	4.7 ± 0.1 ^a^	4.8 ± 0.1 ^a^	4.9 ± 0.2 ^a^
Hepatic injury biomarkers					
Serum aspartate aminotransferase (U/L)	163.6 ± 11.3 ^a^	154.4 ± 23.0 ^a^	152.9 ± 8.1 ^a^	165.8 ± 26.0 ^a^	140.1 ± 20.5 ^a^
Serum alanine aminotransferase (U/L)	60.8 ± 12.1 ^a†^	47.6 ± 6.7 ^b^	50.6 ± 6.3 ^ab^	52.4 ± 7.5 ^ab^	49.8 ± 7.1 ^b^
Serum alkaline phosphatase (U/L)	99.9 ± 16.7 ^a^	91.0 ± 18.3 ^a^	90.3 ± 20.3 ^a^	116.5 ± 30.4 ^a^	92.0 ± 21.1 ^a^

Data are shown as mean ± standard deviation of ten replicates. Different letters indicate significant (*p* < 0.05) differences between samples by Tukey’s or Friedman’s test. ^†^ Indicate significant (*p* < 0.05) difference compared to the HFFD group by Dunnet’s or Wilcoxon test. SD: standard diet; HFFD: high-fat and high-fructose diet; SB: strawberry beverage; BB: blueberry beverage; SBB: strawberry-blueberry beverage; HOMA: homeostatic model assessment; QUICKI: quantitative insulin sensitivity check index: FGIR: fasting glucose-to-insulin ratio; TyG: triglycerides and glucose.

**Table 3 ijerph-20-04418-t003:** Effect of the strawberry, blueberry, and strawberry-blueberry blend beverages on serum, fecal, and hepatic triglycerides, and hepatic fatty acid profile in high-fat and high-fructose diet-fed rats.

Parameters	Control Groups		Beverage Treated Groups		
	SD	HFFD	HFFD + SB	HFFD + BB	HFFD + SBB
Serum triglycerides (mg/dL)	65.44 ± 19.97 ^c†^	160.44 ± 48.70 ^a^	90.13 ± 21.14 ^bc†^	100.44 ± 23.38 ^bc†^	124.20 ± 58.60 ^b†^
Fecal triglycerides (mg/g)	0.14 ± 0.07 ^c†^	0.20 ± 0.05 ^ab^	0.29 ± 0.05 ^a^	0.17 ± 0.07 ^bc^	0.18 ± 0.07 ^bc^
Hepatic triglycerides (mg/g)	23.84 ± 7.24 ^c†^	62.69 ± 15.95 ^a^	38.31 ± 9.30 ^bc†^	38.97 ± 9.47 ^bc†^	45.38 ± 14.07 ^b†^
Hepatic fatty acid profile (mg/g)					
Saturated fatty acids (SFAs)	7.72 ± 1.72 ^c†^	12.79 ± 2.28 ^a^	9.32 ± 2.85 ^bc†^	10.20 ± 2.13 ^abc^	11.86 ± 2.28 ^ab^
Tetradecanoic acid (C14:0)	0.11 ± 0.02 ^b†^	0.25 ± 0.03 ^a^	0.15 ± 0.02 ^ab†^	0.19 ± 0.03 ^ab†^	0.21 ± 0.05 ^a†^
Hexadecanoic acid (C16:0)	3.58 ± 0.66 ^c†^	7.05 ± 1.44 ^a^	5.18 ± 1.82 ^bc†^	5.63 ± 0.53 ^ab^	6.69 ± 1.48 ^ab^
Octadecanoic acid (C18:0)	4.04 ± 1.12 ^b†^	5.57 ± 0.95 ^a^	4.03 ± 1.18 ^ab†^	5.03 ± 0.92 ^ab^	4.98 ± 0.91 ^ab^
Monounsaturated fatty acids (MUFAs)	2.20 ± 0.94 ^b†^	8.49 ± 3.89 ^a^	6.21 ± 3.91 ^ab^	6.53 ± 2.64 ^a^	8.38 ± 3.92 ^a^
*cis*-9-Hexadecenoic acid (C16:1ω7)	0.04 ± 0.02 ^b†^	0.13 ± 0.05 ^a^	0.10 ± 0.05 ^ab^	0.09 ± 0.03 ^a^	0.11 ± 0.04 ^a^
*cis*-9-Octadecenoic acid (C18:1ω9)	2.40 ± 0.55 ^b†^	9.29 ± 2.71 ^a^	6.88 ± 3.36 ^ab^	7.24 ± 1.19 ^a^	9.19 ± 2.74 ^a^
Polyunsaturated fatty acids (PUFAs)	13.54 ± 5.82 ^a^	20.38 ± 7.08 ^a^	15.03 ± 7.72 ^a^	17.68 ± 6.68 ^a^	19.90 ± 6.85 ^a^
*cis*-9,12-Octadecadienoic acid (C18:2ω6)	3.50 ± 0.88 ^b†^	6.90 ± 1.70 ^a^	5.59 ± 2.33 ^ab^	6.01 ± 0.64 ^ab^	7.51 ± 1.83 ^a^
*cis*-9,12,15-Octadecatrienoic acid (C18:3ω3)	0.09 ± 0.03 ^a†^	0.17 ± 0.06 ^a^	0.13 ± 0.08 ^a^	0.14 ± 0.04 ^a^	0.17 ± 0.07 ^a^
*cis*-8,11,14-Eicosatrienoic acid (C20:3ω6)	0.34 ± 0.08 ^a†^	0.55 ± 0.19 ^a^	0.40 ± 0.21 ^a^	0.52 ± 0.10 ^a^	0.54 ± 0.17 ^a^
*cis*-5,8,11,14-Eicosatetraenoic acid (C20:4ω6)	6.97 ± 1.80 ^a^	9.18 ± 1.59 ^a^	6.96 ± 2.24 ^a^	8.59 ± 1.43 ^a^	9.06 ± 1.64 ^a^
*cis*-5,8,11,14,17-Eicosapentaenoic acid (C20:5ω3)	0.21 ± 0.10 ^a^	0.14 ± 0.04 ^ab^	0.12 ± 0.06 ^b^	0.12 ± 0.04 ^b^	0.15 ± 0.05 ^ab^
*cis*-4,7,10,13,16,19-Docosahexaenoic acid (C22:6ω3)	3.93 ± 0.83 ^b†^	5.70 ± 1.12 ^a^	3.70 ± 1.51 ^b†^	4.92 ± 0.95 ^ab^	4.69 ± 1.14 ^ab^
PUFA/SFA index	1.94 ± 0.04 ^a†^	1.73 ± 0.25 ^a^	1.78 ± 0.11 ^a^	1.99 ± 0.42 ^a^	1.85 ± 0.20 ^a^
MUFA/SFA index	0.32 ± 0.05 ^b†^	0.68 ± 0.26 ^a^	0.72 ± 0.14 ^a^	0.61 ± 0.23 ^a^	0.77 ± 0.13 ^a^
MUFA/PUFA index	0.16 ± 0.03 ^b†^	0.38 ± 0.14 ^a^	0.41 ± 0.08 ^a^	0.33 ± 0.13 ^a^	0.42 ± 0.09 ^a^
ω6/ω3 index	2.51 ± 0.15 ^c†^	2.88 ± 0.21 ^b^	3.38 ± 0.26 ^a†^	3.01 ± 0.27 ^b^	3.19 ± 0.18 ^ab†^

Data are shown as mean ± standard deviation of ten replicates. Different letters indicate significant (*p* < 0.05) differences between samples by Tukey’s or Friedman’s test. ^†^ Indicate significant (*p* < 0.05) difference as compared to the HFFD group by Dunnett’s or Wilcoxon test. SD: standard diet; HFFD: high-fat and high-fructose diet; SB: strawberry beverage; BB: blueberry beverage; SBB: strawberry-blueberry beverage.

## Data Availability

The data presented in this study are available on request from the corresponding authors.

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
