# Peer review of "Strawberry, Blueberry, and Strawberry-Blueberry Blend Beverages Prevent Hepatic Steatosis in Obese Rats by Modulating Key Genes Involved in Lipid Metabolism"

_ijerph, 2023, doi:10.3390/ijerph20054418_

Round 1

Reviewer 1 Report

The present study evaluate the effect of strawberry-, blueberry, and strawberry-blueberry blend decoction-based functional beverages on obesity-related metabolic alterations in high fat and fructose diet-fed rats. The results showed that the administration of the three berry-based beverages for eighteen weeks prevented the development of hypertriglyceridemia in obese rats as well as hepatic triglyceride accumulation, and the possible biological mechanism was also explored. The topic is interesting, the paper is well written, the results are described accurately, and the discussion is detailed. The authors should carefully revise their manuscript, taking into account the following:

1.       The research progress of strawberry and blueberry on improving glucose and lipid metabolism needs to be emphasized in the introduction section. What is the difference between current research and previous research? Why should we carry out the new research?

2.       BMI is an indicator of anthropometry in human. Although the author has quoted relevant references, is it authoritative?

3.       Whether the taste of blueberry and strawberry drinks will affect the diet intake of rats? Since the author has monitored the consumption of drinks and diet, it is recommended to add relevant data.

4.       Although both blueberries and strawberries have the effect of reducing triglycerides, it seems that the combination of the two does not have such a strong effect on reducing lipids. What is the possible reason? The author can explain it in the discussion section.

Author Response

The present study evaluates the effect of strawberry-, blueberry, and strawberry-blueberry blend decoction-based functional beverages on obesity-related metabolic alterations in high fat and fructose diet-fed rats. The results showed that the administration of the three berry-based beverages for eighteen weeks prevented the development of hypertriglyceridemia in obese rats as well as hepatic triglyceride accumulation, and the possible biological mechanism was also explored. The topic is interesting, the paper is well written, the results are described accurately, and the discussion is detailed. The authors should carefully revise their manuscript, taking into account the following comments.

R. We appreciate the comment of the reviewer. We provide a point-by-point response to each comment made by the reviewer to improve the quality of our manuscript.

1. The research progress of strawberry and blueberry on improving glucose and lipid metabolism needs to be emphasized in the introduction section. What is the difference between current research and previous research? Why should we carry out the new research?

R.  As suggested by the reviewer, we complemented the introduction section with previous studies carried out with strawberry and blueberry fruits and beverages (Page 2, Lines 56-88). The following references were included in the reference section (Page 19, Lines 718-722).

Quitete, F.T.; Santos, G.M.A.; de Oliveira Ribeiro, L.; da Costa, C.A.; Freitas, S.P.; da Matta, V.M.; Daleprane, J.B. Phenolic-rich smoothie consumption ameliorates non-alcoholic fatty liver disease in obesity mice by increasing antioxidant response. Chem. Biol. Interact. 2021, 336, 109369.

Gonçalves, A.C.; Nunes, A.R.; Flores-Félix, J.D.; Alves, G.; Silva, L.R. Cherries and blueberries-based beverages: Functional foods with antidiabetic and immune booster properties. Molecules 2022, 27, 3294.

2. BMI is an indicator of anthropometry in human. Although the author has quoted relevant references, is it authoritative?

R.  Novelli et al. [21] proposed the assessment of anthropometrical parameters in rats to estimate the development in obesity, which was further associated with the development of an oxidative stress and dyslipidemic state as observed in humans. This reference was included in the methodology section (Page 4, Line 161-163) and the reference section (Page 19, Lines 723-724) to further support this assessment. In addition, we propose to modify the term anthropometric to biometric.

Novelli, E.L.B., Diniz, Y.S., Galhardi, C.M., Ebaid, G.M.X., Rodrigues, H.G., Mani, F., Fernandes, A.A.H, Cicogna, A.C, Novelli-Filho, J.L.V.B. Anthropometrical parameters and markers of obesity in rats. Lab. Anim. 2007, 41, 111-119.

3. Whether the taste of blueberry and strawberry drinks will affect the diet intake of rats? Since the author has monitored the consumption of drinks and diet, it is recommended to add relevant data.

R.  Indeed, we monitored the daily beverage and diet intake throughout the manuscript. Rats supplemented with the berry-based beverages showed a similar daily diet intake and daily beverage intake as compared to the HFFD control group (22.3-22.0 vs. 21.5 g/day/rat) and (40.7-42.1 vs 41.0 mL/day/rat), respectively. Therefore, we consider that rats were not affected by the taste of the berry-based beverages. This information was included in the manuscript (Page 9, Lines 329-332).

4. Although both blueberries and strawberries have the effect of reducing triglycerides, it seems that the combination of the two does not have such a strong effect on reducing lipids. What is the possible reason? The author can explain it in the discussion section.

R. Indeed, as stated by the reviewer, the SBB showed a lower anti-hypertriglyceridemic and hepatoprotective effect than the individual fruit-based beverages (Table 2), even though the SBB showed a similar urinary polyphenol profile as compared to the BB (Figure 3). In this regard, Azzini et al. reported that polyphenols within supplemented mixtures, such as berry extracts, can interact by exerting synergetic, additive, or antagonistic effects, which are difficult to predict in complex models and diseases. We included this statement in the discussion section (Page 17, Lines 648-654) and a perspective in this regard in the conclusion section (Page 17, Lines 653-654). The following reference was included in the reference section (Page 20, Lines 799-800).

Azzini, E.; Giacometti, J.; Russo, G.L. Antiobesity effects of anthocyanins in preclinical and clinical studies. Oxid. Med. Cell. Longev. 2017, 2017, 2740364.

Reviewer 2 Report

Title: Strawberry, blueberry, and strawberry-blueberry blend beverages prevent hepatic steatosis in obese rats by modulating key genes involved in lipid metabolism

Authors: Ana M. Sotelo-González et al.

I suggest the manuscript could be considered for publication in IJERPH after major revising. 

My comments are as follow:

  1. The mechanism of anti-obesity effect should be further investigated.
  2. The introduction section is poor. Author should summary current station of your research topic.
  3. There are some grammatical mistakes that should be checked and proofread carefully.
  4. HPLC, Mass, NMR analysis data were requested.

Author Response

I suggest the manuscript could be considered for publication in IJERPH after major revising. My comments are as follow.

R. We appreciate the comment of the reviewer. We provide a point-by-point response to each comment made by the reviewer to improve the quality of our manuscript.

1. The mechanism of anti-obesity effect should be further investigated.

R. The supplementation with the three berry-based beverages did not prevent the development of diet-induced diet as observed in no effect on body weight gain, triglycerides accumulation in adipose tissue and relative adipose tissue weight as shown in Table 2; therefore, no mechanisms related to anti-obesity effects were assessed.

2. The introduction section is poor. Author should summary current station of your research topic.

R. As suggested by the reviewer, we complemented the introduction section with previous studies carried out with strawberry and blueberry fruits and beverages (Page 2, Lines 56-88). The following references were included in the reference section (Page 19, Lines 718-722).

Quitete, F.T.; Santos, G.M.A.; de Oliveira Ribeiro, L.; da Costa, C.A.; Freitas, S.P.; da Matta, V.M.; Daleprane, J.B. Phenolic-rich smoothie consumption ameliorates non-alcoholic fatty liver disease in obesity mice by increasing antioxidant response. Chem. Biol. Interact. 2021, 336, 109369.

Gonçalves, A.C.; Nunes, A.R.; Flores-Félix, J.D.; Alves, G.; Silva, L.R. Cherries and blueberries-based beverages: Functional foods with antidiabetic and immune booster properties. Molecules 2022, 27, 3294.

3. There are some grammatical mistakes that should be checked and proofread carefully.

R. English writing was revised throughout the manuscript as indicated by the reviewer.

4. HPLC, Mass, NMR analysis data were requested.

R. The polyphenol profile of the berry-based beverages and the urinary polyphenol metabolites were identified and quantified through UPLC-ESI-QToF MSE, as shown in Table 1 and Table 1S, respectively. Each table includes the following identification parameters: retention time, molecular formula, expected (theoretical) mass, observed (experimental) mass, mass error, and adducts. The mass error cutoff was set at <5 ppm as identification criteria. In addition, the supplementary material includes representative mass spectra of major components, which were acquired at low and high collision energy to target the (pseudo)molecular and fragment ions, respectively (Figure S1).

Reviewer 3 Report

I found that this paper to be overall very well informed, objective and well written, however certain points need to be clarified before the manuscript could be considered for publication.

·         What is the abbreviation of HR In materials and methods section?

·         All p in the manuscript should be in italics and capital.

·         How did you keep the over-all experiment-wide P value at < 0.05 given the large number of planned contrasts.

·         The authors should provide Cat No of kits used in this study.

·         Source of Trizol reagent should be provided.

·         In materials and methods section, there are many primers involved in the real-time PCR-PCR test. It is recommended to present them in the form of a table provided with their size, GenBank Accession Numbers or references.

·         The methodological procedures adopted in the histological analysis are not clear.

·         Please, provide the reference of histological analysis.

·         In Tables, letters indicating statistical significant difference should be superscripts.

·         Fig.1 should be colored.

·         Caption of Fig. 1 is incomprehensible. It is necessary to rewritten the caption for this figure.

Author Response

I found that this paper to be overall very well informed, objective and well written, however certain points need to be clarified before the manuscript could be considered for publication.

R. We appreciate the comment of the reviewer. We provide a point-by-point response to each comment made by the reviewer to improve the quality of our manuscript.

1. What is the abbreviation of HR In materials and methods section?

R.  We apologize for the typing mistake, we intended to say RH (relative humidity). We corrected this mistake (Page 3, Line 136).

2. All p in the manuscript should be in italics and capital.

R.  This modification was done throughout the manuscript, as indicated by the reviewer.

3. How did you keep the over-all experiment-wide P value at < 0.05 given the large number of planned contrasts.

R.  In the description of the results, we highlighted when the supplementation with the berry-based beverages significantly (p<0.05) improved a parameter compared to the HFFD control group; nevertheless, not all parameters showed statistical significance. In the revised version of the manuscript, we included the specific Pvalues of the significant parameters improved by the berry-based beverages.

4. The authors should provide Cat No of kits used in this study.

R.  We included all the catalog numbers in the methodology section as indicated by the reviewer (Page 4, Lines 171, 172, 173, 175, 192, and 199).

5. Source of Trizol reagent should be provided.

R.  Trizol reagent was purchased from Invitrogen was described in the section 2.1. Chemical and reagents (Page 2, Line 93).

6. In materials and methods section, there are many primers involved in the real-time PCR-PCR test. It is recommended to present them in the form of a table provided with their size, GenBank Accession Numbers or references.

R.  The primer sequences and the GenBank accession numbers were included in Table S4 (supplementary material) as suggested by the reviewer.

7. The methodological procedures adopted in the histological analysis are not clear.

R.  The methodology of the histological analysis was rewritten as suggested by the reviewer (Page 5, Lines 243-247).

8. Please, provide the reference of histological analysis.

R.  We included the reference of the histological analysis as indicated by the reviewer (Page 18, Lines 737-738).

Sethunath, D.; Morusu, S.; Tuceryan, M.; Cummings, O.W.; Zhang, H.; Yin, X.M.; Vanderbeck, S.; Chalasani, N.; Gawrieh, S. Automated assessment of steatosis in murine fatty liver. PLoS One 2018, 13, e0197242.

9. In Tables, letters indicating statistically significant difference should be superscripts.

R.  This modification was done in Tables 1, 2, 3, and S1 as suggested by the reviewer.

10. Fig.1 should be colored.

R. Figure 1 is now colored as suggested by the reviewer.

11. Caption of Fig. 1 is incomprehensible. It is necessary to rewritten the caption for this figure.

R.  The caption of Figure 1 was rewritten as follows as suggested by the reviewer (Page 11, Lines 392-395).

Round 2

Reviewer 1 Report

All my review comments have received positive responses.

Reviewer 2 Report

Accepted

Reviewer 3 Report

Now all the comments are addressed and significantly revised. I recommend the paper for publication.